# Public awareness of gastric cancer risk factors and screening behaviours in Shijiazhuang, China: A community-based survey

Qian Wang[1], Xiao-Ci He[1], Lian-Xia Geng[2], Shu-Lin Jiang[3], Chuan-Jie Yang[3], Kai-Yue Xu[4], Shu-Fang Shen[5], Wen-Wen Cao[6], Wei Qi[3]*, Shu-Ping Zhao[3]*

1 Department of Health Management, The Second Hospital of Hebei Medical University, Shijiazhuang, China, 2 Department of Human Resource management, The Second Hospital of Hebei Medical University, Shijiazhuang, China, 3 Department of Gastroenterology, Hebei Key Laboratory of Gastroenterology, Hebei Institute of Gastroenterology, The Second Hospital of Hebei Medical University, Hebei Clinical Research Center for Digestive Diseases, Shijiazhuang, China, 4 Department of Clinical Laboratory, Hebei Hospital of Traditional Chinese Medicine, Affiliated Hospital of Hebei University of Chinese Medicine, Shijiazhuang, China, 5 Department of Pediatrics, The Second Hospital of Hebei Medical University, Shijiazhuang, China, 6 Office of Academic Affairs, The Second Hospital of Hebei Medical University, Shijiazhuang, China

* 315965707@qq.com (S-PZ); 28502620@hebmu.edu.cn (WQ)

**Data Availability Statement:** All relevant data are within the manuscript and its Supporting Information files.

## Abstract

### Background

Reducing exposure to risk factors and screening represent 2 major approaches to gastric cancer (GC) prevention, but public knowledge GC risk factors and screening behaviour remain unknown. We aimed to investigate public awareness of GC risk factors, adherence to screening, and barriers hindering screening practices in China.

### Methods

This community-based household survey was conducted within Shijiazhuang, China, and 1490 residents were recruited through a multistage stratified cluster random sampling approach. A self-administered questionnaire was completed which consisted of three sections: demographics, awareness of GC risk factors, and personal screening behaviours. Factors associated with knowledge of risk factors and screening behaviours were evaluated using binary logistic regression analysis.

### Results

The mean risk factor awareness score of 12 (7, 15) revealed insufficient knowledge in 51.1% of participants. Dietary lifestyle factors were better understood than physical activity and weight-related factors. Marital status (OR 1.967; 95% CI 1.415 to 2.734), higher income (OR 1.197; 95% CI 1.010 to 1.418), and a history of upper gastrointestinal problems (OR 0.048; 95% CI 1.002 to 1.311) were associated with higher awareness. Merely 21.5% underwent GC screening, with higher rates linked to older age (OR 1.642; 95% CI 1.418 to 1.902), higher education (OR 1.398; 95% CI 1.176 to 1.662), a history of upper gastrointestinal problems (OR 3.842; 95% CI 2.833 to 5.209), and moderate (OR 2.077; 95% CI 1.352 to

**Funding:** The author(s) received no specific funding for this work.

**Competing interests:** The authors have declared that no competing interest exsist.

3.191) and high (OR 2.529; 95% CI 1.311 to 4.878) perceived GC risk. Notably, participants commonly refused gastroscopy due to the absence of symptoms or signs.

## Conclusions

In Shijiazhuang, more than half of participants demonstrated inadequate knowledge of GC risk factors, and screening participation rates were remarkably low. This emphasizes the need for targeted interventions to enhance GC awareness and significantly improve screening rates.

## Introduction

Gastric cancer (GC) ranks as the fifth most prevalent cancer globally and is responsible for the third-highest number of cancer-related deaths, with over one million new cases and 769,000 deaths in 2020 [1]. However, the incidence and mortality of gastric cancer are highly variable by region. More than half of all cases occur in East Asia, Eastern Europe, South America and Western Asia [2]. Considerably higher incidence rates are observed in Eastern Asia, particularly China. China accounts for approximately half of the worldwide incident cases and deaths related to GC [3]. Geographic variation in the incidence of GC appeared due to several reasons, and environmental factors, especially unhealthy lifestyle, are the most important risk factors. On the other hand, the high mortality rate due to the fact that more than 80% GC cases are diagnosed at an advanced stage [4]. Therefore, reducing exposure to risk factors and implementing screening programs remain the two primary approaches for alleviating the burden of GC.

Adopting and maintaining healthy behaviours is key to reducing exposure to risk factors. Compelling evidence suggests that adhering to a healthy lifestyle could potentially prevent 59.8% of GC cases and deaths [5], highlighting the pivotal role of healthy behaviours in GC prevention. Similarly, in Singapore, it reduces GC risk by up to 50% among the Chinese population [6]. A healthy lifestyle can also reduce a high genetic risk of GC by 1.12% [7]. Unfortunately, research from many countries shows that adherence in the general population is worryingly low [8–12].

Effective prevention necessitates both collective actions and individual actions [13]. A comprehensive understanding of public awareness regarding GC risk factors significantly influences the development and execution of effective prevention strategies against this disease. For individual actions to occur, public awareness of cancer risk factors is a crucial prerequisite. Unfortunately, public understanding of cancer risk factors remains uneven, despite numerous cancer awareness campaigns in Western countries in recent years. According to the 2019 AICR Cancer Risk Awareness Survey in the United States, 89% recognize smoking as a cancer risk factor, but only 26% are aware of the risks associated with grilling meat. Similarly, in the United Kingdom, 90% recognize smoking risks, but only 30% understand the risks of low fruit and vegetable consumption [14]. Despite China's efforts to raise public awareness, such as the "Healthy China 2030" initiative, there is a paucity of data concerning general public awareness of risk factors for GC in China.

Endoscopic screening has substantially contributed to the decline in GC mortality by enabling the detection and removal of precancerous lesions [15]. Effective screening was related to an approximately 40% reduction in GC mortality risk [16,17], with its success contingent upon public participation [18]. Extensive research in Western countries consistently

demonstrates relatively high rates of public uptake in cancer screening programs, such as the United States (cervical, 72.4%; breast, 75.9%; colorectal, 71.8%) and the United Kingdom (cervical, 71.4%; breast, 71.1%; colorectal, 57.7%) [19,20]. In Asia, Nationwide organized GC screening programs have been running for decades in South Korea and Japan, more than half of GCs are discovered at an early stage [21]. However, in China, the early diagnosis and treatment rate of GC is below 10% [22], and public participation in GC screening remains limited. Moreover, the factors influencing screening behaviours and the perceived barriers to GC screening are largely unknown.

Therefore, the primary objective of this study is to evaluate the knowledge levels of GC risk factors and screening behaviours among the general population in China. We also aim to identify the factors influencing awareness and screening uptake, as well as the perceived barriers preventing individuals from undergoing GC screening. The findings have significant implications for health-related behavior change and may enhance GC prevention and control efforts.

## Material and methods

### Study design and study participants

This community-based household survey study was conducted in Shijiazhuang, the provincial capital of Hebei province in northern mainland China, with a registered population of approximately 11.2 million as of 2022. Shijiazhuang's urban area is divided into five administrative districts. Employing a multistage stratified cluster random sampling approach, two streets were randomly chosen from each district, followed by the random selection of two neighborhood committees from each street. Adhering to simple random sampling principles, 80 households were selected from each committee. The KISH grid sampling method was then employed to select one resident meeting predefined criteria from each household for the survey. In total, 1,600 participants were selected, meeting eligibility criteria of being Chinese, aged over 18 years, and residing in Shijiazhuang. Exclusions comprised individuals unable to understand Chinese, those with a history of cancer, those declining participation, and those too ill to provide consent.

After obtaining written informed consent, each participant was asked to complete a face-to-face interview and questionnaire regarding their knowledge of gastric cancer risk factors and screening behaviours. This study was approved by the Ethics Committee of the Second Hospital of Hebei Medical University (Grant number: 2022-R137). All personally identifiable information was removed before data analysis. The study adhered to the STROBE (Strengthening the Reporting of Observational Studies in Epidemiology) statement.

### Questionnaire

This questionnaire was designed based on expert discussion and relevant published articles [2,5,7,23–25]. The panel of experts involved in this process comprised four digestive disease specialists, a health management specialist, a social work specialist, and an epidemiologist. All questions were reviewed for content, understanding, and comprehensibility. Prior to data collection, a pilot test of the questionnaire was conducted among 100 individuals aged ≥18 years over in downtown Shijiazhuang to assess its suitability and feasibility.

The questionnaire included:

Demographic and personal characteristics, including age, sex, marital status, educational level, monthly income, family history of GC, perceived risk of GC, and the residents themselves, family members, or close friends experienced upper gastrointestinal (GI) problems.

A questionnaire comprising 18 items was used to assess the knowledge of GC risk factors. These items were selected based on two criteria: 1) the presence of sufficient evidence indicating a probable causal relationship with GC and 2) the availability of risk factor data from the

China Health and Nutrition Survey. The identified risk factors included smoking, excessive alcohol consumption, physical inactivity, high sodium intake, high red meat consumption, pickled vegetable consumption, low fruit and vegetable intake, frequent consumption of leftovers, fast eating habits, overeating, consumption of smoked foods, irregular diet patterns, stress, H. pylori infection, male sex, older age, family history of GC, and stomach diseases. Participants responded to each item using a Likert scale, where 'strongly agree' and 'agree' responses were combined and scored as '1'; while other responses such as 'not sure', 'disagree', and 'strongly disagree' were scored as '0'. The total scores ranged from zero to 18. As an indicator of "good" knowledge, a cut-off level of 70% of individual percentage scores was used [26].

Endoscopic screening behaviour and barrier assessment: (i) Participants were asked the question, 'Have you ever undergone GC screening?' with response options of 'yes' or 'no'. (ii) For participants who had not undergone GC screening, they were asked to indicate the reasons for their non-participation. Response options included 'no symptoms or signs', 'procedural pain and discomfort', 'fear of being cheated', 'fatalistic beliefs about GC' 'worried about screening results', 'lack of time', 'financial limitations', and 'difficulty in making appointments'. Multiple answers were allowed for this question.

## Data collection

Questionnaire data were collected through face-to-face written interviews conducted among 1,600 residents in Shijiazhuang. The interviews were conducted by a team of well-trained interviewers, comprising eight nurses from the Second Hospital of Hebei Medical University. Before conducting the interviews, the team members underwent a minimum of 3 days of training on obtaining consent and administering the questionnaire. They were required to pass an assessment before being authorised to interview participants. This study was performed between September and November 2022. During the interviews, participants were encouraged to complete the questionnaire independently to the best of their abilities. However, for those who required assistance, the interviewers provided support for them. Each interview lasted for approximately 10 min. As a token of appreciation, each participant received a gift valued at ¥20 (equivalent to approximately US $3).

## Statistical analyses

Data analysis was performed using SPSS version 24 (SPSS, Inc., Chicago, IL, USA) software. Normally distributed continuous data (according to the Shapiro–Wilk test) are presented as means ± standard deviations, and those with skewed distributions are expressed as medians (interquartile range). Categorical variables are expressed as numbers and percentages. For a significant relationship, binary logistic regression analysis was performed to identify factors that influence the participants' knowledge and screening behaviour. All statistical tests were two-sided, and $p < 0.05$ was considered statistically significant.

## Results

### Demographic and personal characteristics of the participants

A total of 1,600 eligible participants were recruited and 1,502 completed the survey, with a response rate of 93.88%. 12 questionnaires were excluded from analysis due to incomplete information, leaving 1490 valid for further analysis. Demographic characteristics of the respondents are presented in Table 1, indicating a higher representation of females, individuals aged over 60 years, married individuals, individuals with medium education levels, and individuals with low incomes. Among the respondents, 24.5% reported personal experience or had

**Table 1. Demographic characteristics of survey participants.**

| Parameter | Number and Percentage of Participants—n (%) |
|---|---|
| **Sex** | |
| male | 522 (35.0) |
| female | 968 (65.0) |
| **Age** | |
| 18~29 | 84 (5.6) |
| 30~39 | 353 (23.7) |
| 40~49 | 221 (14.8) |
| 50~59 | 381 (25.6) |
| ≥60 | 451 (30.3) |
| **Marital status** | |
| unmarried | 279 (18.7) |
| married | 1 185 (79.6) |
| divorce | 11 (0.7) |
| widowed | 15 (1.0) |
| **Educational level** | |
| Uneducated or elementary school | 176 (11.8) |
| Middle school | 428 (28.7) |
| High school | 430 (28.9) |
| College | 422 (28.3) |
| Master degree or above | 34 (2.3) |
| **Income[a]** | |
| <¥3 000 | 857 (57.5) |
| ¥3 000~5 000 | 431 (28.9) |
| ¥5 000~10 000 | 161 (10.8) |
| >¥10 000 | 41 (2.8) |
| **You, family members or friends ever suffer upper gastrointestinal problems** | |
| yes | 365 (24.5) |
| no | 1 125 (75.5) |
| **Family history of gastric cancer** | |
| yes | 140 (9.4) |
| no | 1 350 (90.6) |
| **Perceived risk of gastric cancer** | |
| Low | 760 (51.0) |
| Moderate | 153 (10.3) |
| High | 55 (3.7) |
| Do not know | 522 (35.0) |

**Notes:** Family history of gastric cancer included history of gastric cancer in parents, children, or Siblings. [a] Income presented in Chinese Yuan (¥).

a family member/friend with experience of an upper GI problem, 9.4% had a family history of GC, and only 3.7% of residents had a high risk perception for GC.

## Knowledge about risk factors for GC

Fig 1 shows the proportion of correct responses to all 18 knowledge items. The 18 items for knowledge scores had a reliability (Cronbach's α) of 0.929. The overall mean awareness score

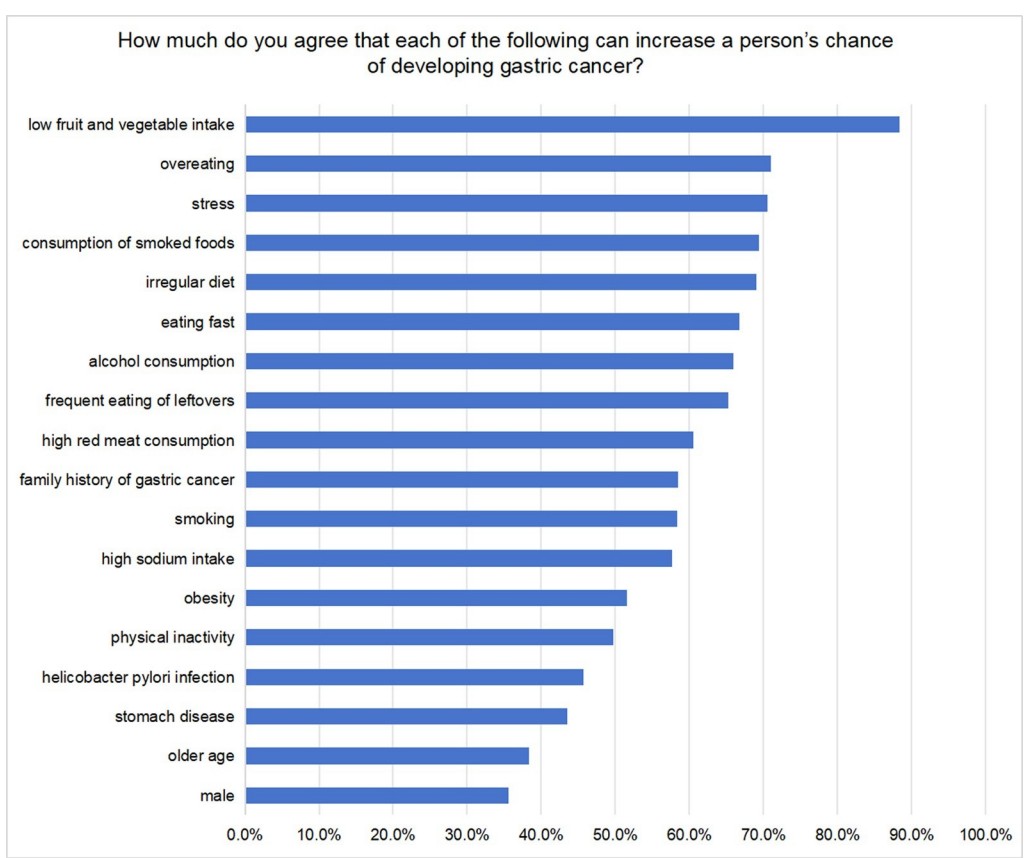

**Fig 1. Proportion of participants who agreed or strongly agreed that a given risk factor increased a person's chance of developing gastric cancer.**

for GC risk factors was reported as 12 (7, 15), and 48.9% of residents obtained a score of ≥70%. This indicated that more than half of the participants had inadequate knowledge regarding GC risk factors. Among the specific risk factors, participants demonstrated the highest awareness of low fruit and vegetable intake (88.3%), overeating (71%), and stress (70.5%). Conversely, they were least aware of male (35.6%), older age (38.3%), and stomach diseases (53.5%). The pattern of awareness, with better recognition of dietary lifestyle factors compared with physical activity and weight, aligns with findings from previous studies [18,27,28].

Table 2 presents the analysis of the factors determining the knowledge of GC risk factors. The results indicate that married individuals (odds ratio 1.967; 95% confidence interval [CI] 1.415 to 2.734) exhibited a higher knowledge level compared with unmarried individuals. Additionally, participants with a higher monthly income (OR 1.197; 95% CI 1.010 to 1.418) demonstrated a higher knowledge score. Moreover, individuals with a history of upper GI problems (OR 0.048; 95% CI 1.002 to 1.311) exhibited a higher level of knowledge regarding risk factors for GC.

## Screening behaviour

Of all the participants, only 320 (21.5%) individuals underwent GC screening, indicating a significant portion of participants (78.5%) had not participated in GC screening.

Logistic regression analysis showed that several variables were significantly associated with GC screening behaviour, including education, age, and experience of upper GI problems ($p < 0.05$).

**Table 2. Multivariate binary logistic regression analysis of factors associated with awareness of gastric cancer risk factors (n = 1490).**

| Variable | B | SE | wald | P | OR (95% CI) |
|---|---|---|---|---|---|
| marital status | | | | | |
| married | .676 | .168 | 16.223 | <0.001 | 1.967 (1.415 to 2.734) |
| divorced | .664 | .635 | 1.095 | .295 | 1.943 (0.560 to 6.744) |
| widowed | 1.045 | .579 | 3.251 | .071 | 2.843 (0.913 to 8.850) |
| unmarried | | | 1 (ref) | | |
| income | .180 | .087 | 4.328 | .037 | 1.197 (1.010 to 1.418) |
| experience of upper gastrointestinal problems | | | | | |
| yes | .271 | .137 | 3.896 | | 0.048 (1.002 to 1.311) |
| no | | | 1 (ref) | | |

**Abbreviations:** OR, odds ratio; CI, confidence interval; ref, reference.

Older age (OR 1.642; 95% CI 1.418 to 1.902) and higher levels of education (OR 1.398; 95% CI 1.176 to 1.662) were positively correlated with higher rates of GC screening. Additionally, residents with a history of upper GI problems (OR 3.842; 95% CI 2.833 to 5.209) were more likely to have undergone GC screening compared with those without such experiences. Furthermore, individuals with moderate (OR 2.077; 95% CI 1.352 to 3.191) and high (OR 2.529; 95% CI 1.311 to 4.878) levels of perceived risk for GC were more inclined to undergo GC screening (Table 3).

Among participants who did not undergo GC screening (n = 1170), perceived barriers to screening were examined. The most common barrier was the 'absence of symptoms or signs' (85.3%), followed by 'discomfort during performance' (16.0%), and the 'lack of time' (10.8%) (Fig 2).

## Discussion

In the present study, we observed that more than half of participants demonstrated inadequate knowledge regarding GC risk factors, and most residents had not undergone GC screening. Furthermore, our findings revealed that the primary barriers to screening participation were the absence of symptoms or signs, concerns about procedural pain and discomfort, and time constraints.

### Knowledge of GC risk factors

This study indicated that 48.9% of survey participants had a good knowledge level regarding GC risk factors. It was similar to a recent study in other regions of China (47.5%) [28], but lower than the results from the 2020 Spanish Onco-barometer survey (76.7%) [29].

Knowledge regarding dietary lifestyle factors was better than knowledge about the impact of physical activity or weight. Similar trends were noted globally; a study across 20 countries revealed that awareness of physical inactivity and overweight was lower compared to other lifestyle factors, with only 28% and 29%, respectively [30]. In the European colorectal cancer risk awareness study, 70% were familiar with dietary factors, but only 30% recognized the link between lack of exercise and cancer risk [27]. Results from a recent Chinese study on public awareness of gastric cancer indicated a lower knowledge level (54%) concerning obesity [28]. A worrying finding is that obesity is projected to become the primary risk factor for GI cancer in the coming years in China, with an estimated contribution of over 141,000 new GI cancer cases by 2031 [5]. Additionally, a study by Ma et al. highlights the alarming rise in obesity rates in China, particularly abdominal obesity [31]. The growing burden of obesity could be driven

**Table 3. Multivariate binary logistic regression analysis of factors associated with gastric cancer screening behavior (n = 1490).**

| variable | B | SE | wald | P | OR (95% CI) |
|---|---|---|---|---|---|
| age | .496 | .075 | 43.743 | .000 | 1.642 (1.418 to 1.902) |
| educational level | .335 | .088 | 14.429 | .000 | 1.398 (1.176 to 1.662) |
| experience of upper gastrointestinal problems | | | | | |
| yes | 1.346 | .155 | 74.991 | .000 | 3.842 (2.833 to 5.209) |
| no | | | 1 (ref) | | |
| perceived risk of GC | | | | | |
| moderate | .731 | .219 | 11.120 | .001 | 2.077 (1.352 to 3.191) |
| high | .928 | .335 | 7.657 | .006 | 2.529 (1.311 to 4.878) |
| do not know | .126 | .154 | .673 | .412 | 1.135 (0.839 to 1.534) |
| low | | | 1 (ref) | | |

**Abbreviations:** OR, odds ratio; CI, confidence interval; ref, reference; GC, gastric cancer.

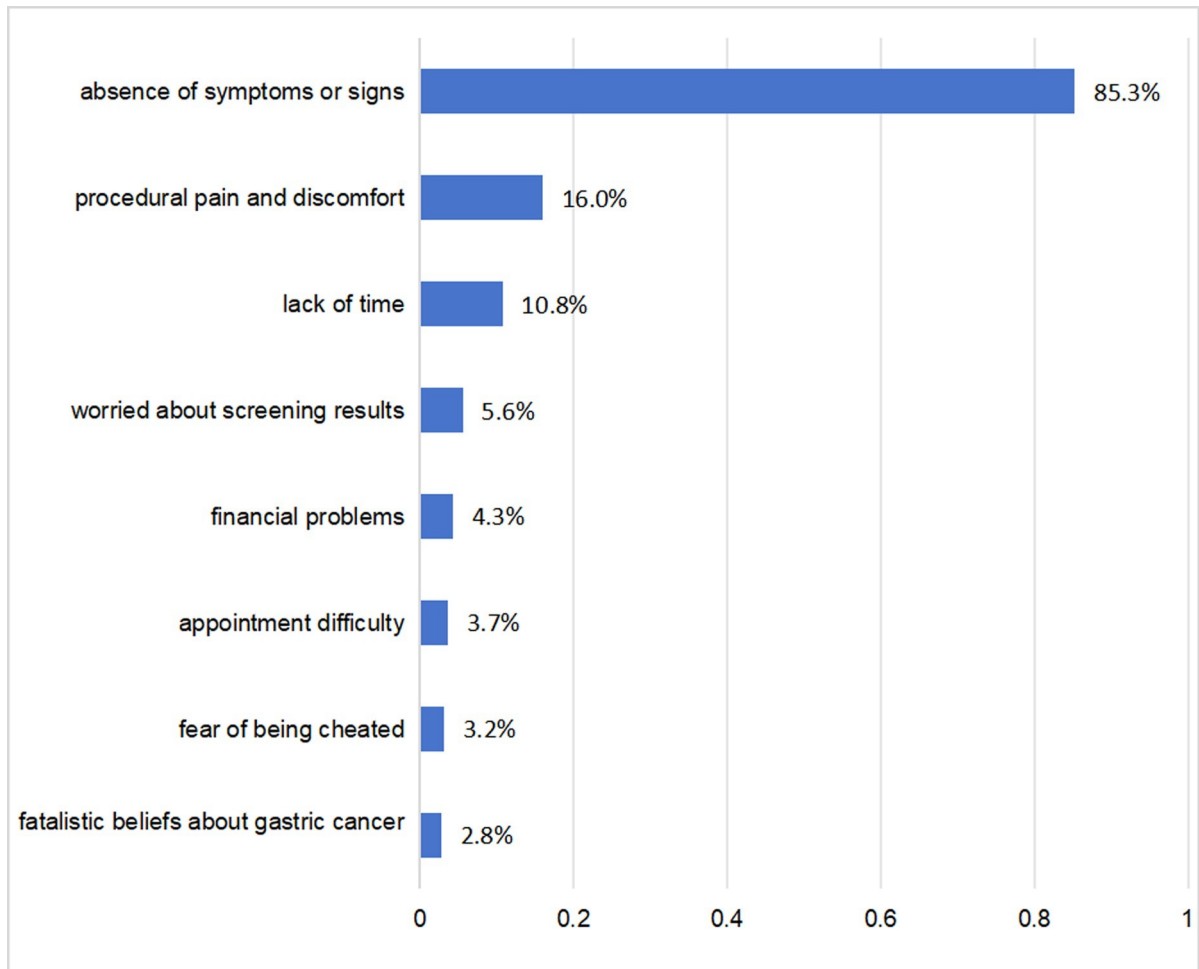

**Fig 2. Perceived barriers towards gastric cancer screening.**

by physical inactivity and dietary factors, such as increased consumption of animal-source foods, refined grains, and highly processed, high-sugar, and high-fat foods [32]. Therefore, an effective method is required for governments, social media, professionals and the community to work together to improve cancer awareness and modify unhealthy lifestyles. Leveraging the internet to disseminate professional and authoritative prevention knowledge can effectively promote a healthy lifestyle and foster a supportive environment for it. By enhancing public awareness, compliance with health behaviors will subsequently improve, enabling more individuals to adopt and maintain practices that are beneficial to their well-being.

Besides modifiable risk factors awareness, age and sex were the two least known non-modifiable risk factors, with only 38.3% and 35.6%, respectively. This finding is consistent with a study by Kyle et al., where they reported that 35.6% of individuals identified older age and 34.7% identified male sex as risk factors for cancer [33]. These results indicate that people tend to be easier to neglect on-modifiable factors. Raising awareness of these risk factors among residents may encourage their participation in cancer screening programs. Furthermore, our study revealed that participants had limited knowledge of disease risk factors (43.5%). This finding suggests that individuals are unfamiliar with these medical terms, indicating a potential gap in educational interventions.

Through the binary logistic regression model, associations were identified between marital status, income, and family history of GC with GC risk factors. Married individuals demonstrated better knowledge of these risk factors compared with non-married individuals. This finding suggests that spouses might learn from each other and have increased access to health information. A similar result was reported by Maram et al. [34] in their survey on public awareness of colorectal cancer in Riyadh, where married respondents provided more correct answers than single respondents. Moreover, participants with a family history of GC exhibited a higher knowledge level of GC risk factors. This could be attributed to their increased exposure to discussions about GC with their families. Additionally, higher income was associated with greater awareness of GC risk factors. These findings align with previous studies that have shown individuals with higher socioeconomic status to be more aware of cancer risk factors and protective behaviours [35,36]. This underscores the importance of targeting residents with lower socioeconomic status as a key population, especially since there's a prevalent myth among them that being diagnosed with cancer equates to a death sentence and that cancer is an individual's fate and not preventable [37]. Focusing on comprehensive education—from the individual level to communities, governments, and beyond—is vital to debunking these misconceptions and enhancing awareness about cancer prevention and treatment options.

## Screening behaviour

Screening is the cornerstone of CRC prevention, and gastroscopy is considered by many to be the gold standard for GC screening. However, the compliance rate for endoscopic screening in China is only 18.41% [38]. In our study, as anticipated, a small proportion of participants (21.5%) underwent endoscopic screening at least once. Although this percentage was marginally higher than that reported in another study conducted in China (15.2%) [39], it remains lower than the rates reported in Korean (72.8%) [40] and Japanese (36.7%) [41] studies.

Higher participation rates in GC screening were observed among individuals of older age, who had higher education levels, perceived higher risk of GC, and experienced upper GI problems.

The perceived susceptibility to GC was generally low, with only 3.7% of the participants perceiving themselves to be at high risk. This finding aligns with previous studies [42]. A greater perceived risk of a disease can sometimes be associated with more positive preventive

behaviour. In our study, it was also observed that individuals with a higher perceived risk were significantly more likely to participate in GC screening. Similar findings have been reported in previous studies. For instance, a Spanish study conducted among a family-risk population reported that a high subjective perception of risk independently predicted colorectal cancer screening participation (OR 2.87; 95% CI 1.10–7.46; p = 0.03) [43]. Another study in the United Kingdom revealed that participants who believed their risk was higher than that of the average-risk population were more willing to participate in colorectal cancer screening (98%) compared with those who believed their risk (84%) was the same [44]. However, it is important to note that while perceived risk plays a significant role in influencing cancer screening behaviour, it is not an independent predictor of its own. Other mediating factors are required to affect screening behaviour.

It was observed that older participants were more likely to undergo GC screening than younger participants. This trend aligns with observations made in studies on other types of cancer screening. In Italy, colorectal screening uptake was higher among elderly (≥65 years) individuals compared with younger invitees [45]. In England, in 2018–2019, there was a 68.2% uptake of breast cancer screening among women aged 50–52 years, which increased to 73.2% among those aged 65–70 years [46]. Furthermore, the uptake of lung cancer screening also tends to be higher among older age groups [47]. This pattern might be attributed to factors such as time constraints and less concern about cancer risk among younger individuals.

Apart from age, the findings revealed a significant association between higher educational levels and increased uptake of GC screening. This finding aligns with existing evidence indicating that individuals with at least 12 years of education are more likely to use cancer-screening services compared with individuals who have fewer years of formal education [48]. Our findings suggest that individuals with higher educational attainment might possess greater health consciousness and have easier access to screening facilities compared with individuals with lower levels of institutional education. Educational level affects cancer awareness, screening uptake and behavioural risk. Essentially, those most in need of screenings are often the least informed about cancer risks and the benefits of early detection.

Our study revealed a strong relationship between individuals who had family members or close friends with a history of upper GI problems and their screening behaviour. This association has been reported in other studies conducted among Asian populations [39,49]. The onset of uncomfortable symptoms might serve as a motivating factor to seek medical help and participate in screening.

Perceived barriers have a major impact on screening uptake. Understanding and identifying the barriers to screening uptake are essential for professionals to plan effective interventions. In the present study, the 'absence of symptoms or signs' was the primary reason why participants refused to undergo gastroscopy, which is consistent with findings from a previous study conducted in China [39]. This finding implied that most respondents were unaware that screening is intended for asymptomatic individuals. Consequently, they might delay seeking medical advice until symptoms are present, leading to a more advanced stage of the disease. Therefore, efforts must be focused on dispelling misconception among the general population. Another significant perceived barrier identified in our study was the fear of pain or discomfort during the screening procedure. A survey reported that although 83.8% of individuals visiting medical centres recognised the usefulness of GC screening, only 15.2% underwent the procedure, while 38.1% experiencing fear of gastroscopy [39]. Therefore, there is a pressing need to develop noninvasive methods that could complement and enhance endoscopic screening. Novel, accurate, and reliable biomarkers that are highly integrated could represent a significant advancement in this field. This also presents a considerable challenge for gastroenterologists,

who are tasked with the critical role of integrating these innovative tools into current screening practices.

Our findings have important clinical and public health implications. First, we provide valuable reference data to help decision-makers enhance cancer awareness and screening rates. It is advisable for them to establish an authoritative platform for science communication, to compile and disseminate core information and key points on cancer prevention and treatment. Simultaneously, they should optimize cancer screening management and establish a tiered screening system. Secondly, professionals will tailor health education initiatives to address public misconceptions and knowledge gaps, thereby tackling inequalities in awareness. Additionally, barriers to screening motivate doctors to develop novel, non-invasive, and precise biomarkers, which further advance screening technologies. Third, although raising public awareness is a relatively gentle method for achieving behavioral change, it is immensely beneficial for enhancing the overall health literacy of the population and encouraging long-term adherence to healthy behaviors. Better understanding of risk factors and increased prevention efforts offer cost-effective and sustainable means to reduce the global cancer burden in the longer term.

### Strength and limitations

The present study possesses several notable strengths. First, the samples were drawn from naturally formed communities, enhancing the generalisability of our findings to the overall population. Additionally, a detailed and comprehensive questionnaire was used to assess knowledge regarding risk factors for GC. Moreover, the response rate of this study was 93.88%, which might be attributed to the use of face-to-face interviews. However, the present study also has several limitations. First, self-reported questionnaires were employed, which inherently carry the risk of bias, including potential overestimation or underestimation due to social desirability. Second, Chinese traditional cultural norms and beliefs that might also influence an individual to participate in cancer screening have not been discussed. Third, the lack of a validated assessment tool for GC-related knowledge, preferences, attitudes, and behaviour is a limitation in the present study and previous studies on this subject matter.

### Conclusion

In this study, we found that more than half of participants demonstrated a significant gap in knowledge regarding cancer risk factors, and concurrently, the rate of public engagement in GC screening was alarmingly low. Urgent development of a cancer screening strategy tailored to the Chinese population, coupled with educational initiatives to raise awareness and dispel misconceptions, is imperative. Future research should explore the correlation between risk-reduction behavior and self-perceived knowledge of GC risk factors. Additionally, understanding the intricate interplay between knowledge, attitudes, and various factors affecting screening behavior is essential for optimizing cancer screening services and reducing the disease burden.

### Supporting information

**S1 Raw data.**
(XLSX)

### Acknowledgments

The authors gratefully acknowledge the participants who participated in this study. We would like to thank Mr. Jing Dai for providing statistical consultation. In particular, the advice of Mr.

Kangwei Xun (Living with Disability Research Centre, La Trobe University, Bundoora, Victoria, Australia) on manuscript writing is sincerely appreciated. We are especially grateful to Professor Mingzi Li (Peking University Health Science Center, Beijing, China) for her critical guidance during the revision of this manuscript.

## Author Contributions

**Conceptualization:** Qian Wang, Shu-Lin Jiang, Shu-Ping Zhao.

**Data curation:** Xiao-Ci He, Shu-Fang Shen.

**Formal analysis:** Xiao-Ci He, Kai-Yue Xu, Wei Qi.

**Funding acquisition:** Shu-Ping Zhao.

**Investigation:** Kai-Yue Xu, Shu-Fang Shen, Wen-Wen Cao.

**Methodology:** Qian Wang, Xiao-Ci He, Shu-Lin Jiang, Wei Qi.

**Project administration:** Chuan-Jie Yang, Shu-Ping Zhao.

**Resources:** Lian-Xia Geng, Shu-Ping Zhao.

**Software:** Xiao-Ci He.

**Supervision:** Qian Wang, Lian-Xia Geng, Chuan-Jie Yang, Wei Qi.

**Validation:** Wei Qi.

**Writing – original draft:** Qian Wang, Shu-Ping Zhao.

**Writing – review & editing:** Qian Wang, Wei Qi, Shu-Ping Zhao.

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
