## [Decision Letter · Decision Letter 0]

22 May 2024

PONE-D-23-38363Public Awareness of Gastric Cancer Risk Factors and Screening Behaviours in Shijiazhuang, China: A Community-based SurveyPLOS ONE

Dear Dr. zhao,

Thank you for submitting your manuscript to PLOS ONE. After careful consideration, we feel that it has merit but does not fully meet PLOS ONE’s publication criteria as it currently stands. Therefore, we invite you to submit a re

We look forward to receiving your revised manuscript.

Kind regards,

Peivand Bastani

Academic Editor

PLOS ONE

Journal Requirements:

5. Please amend the manuscript submission data (via Edit Submission) to include author Lianxia Geng, Shulin Jiang, Chuanjie Yang, Kaiyue Xu, Shufang Shen and Wenwen Cao.

Reviewers' comments:

Reviewer's Responses to Questions

**Comments to the Author**

1. Is the manuscript technically sound, and do the data support the conclusions?

Reviewer #1: Yes

Reviewer #2: Yes

2. Has the statistical analysis been performed appropriately and rigorously? 

Reviewer #1: Yes

Reviewer #2: Yes

3. Have the authors made all data underlying the findings in their manuscript fully available?

Reviewer #1: Yes

Reviewer #2: Yes

4. Is the manuscript presented in an intelligible fashion and written in standard English?

Reviewer #1: Yes

Reviewer #2: Yes

5. Review Comments to the Author

Reviewer #1: Wang Q et al submit their study regarding the Public Awareness of Gastric Cancer Risk Factors and Screening

2 Behaviours in Shijiazhuang, China: A Community-based Survey.

Given the high incidence of GC in China, it is an important public health issue that needs multiple levels of data gathering and analysis and evaluating the public awareness is a good part of it.

The study is well designed and analyzed.

It seems that the main interest of authors is to find the link between public knowledge and screening. It could be good to emphasize also how improving public knowledge could lead to change of lifestyle and acting as a tertiary preventive measure.

Figures look like being screenshot with low resolution. It is important to have high quality images for the final publication.

Reviewer #2: The article is of excellent quality. It is a very important topic for the scientific community. The small points I suggest improving are:

- Present an international overview of the theme in the introduction (what was done similarly in other countries, or even what was not done);

- Improve the practical implications of the study (How important is the study for society? How important is the study for professionals? How important is the study for the health system? Etc..

6. PLOS authors have the option to publish the peer review history of their article (what does this mean?). If published, this will include your full peer review and any attached files.

Reviewer #1: **Yes: **Behnoush Abedi-Ardekani

Reviewer #2: **Yes: **Mateus Antunes

---

## [Author Response · Author response to Decision Letter 0]

23 Jul 2024

Dear Dr. Bastani,

Thank you for the thorough review of our manuscript and the detailed feedback provided. We appreciate the opportunity to address these comments and submit a revised version of our paper.Below, we outline the changes made in response to the specific points raised.

Point 1: PLOS ONE Style Requirements

Reviewer Comment: Please ensure that your manuscript meets PLOS ONE's style requirements, including those for file naming.

Response: We have revised our manuscript to meet PLOS ONE's style requirements, including modifications to the title page, abstract, main text, tables, and captions as per the guidelines. Additionally, we have added an acknowledgment section to express our gratitude to those who contributed to the study.

Point 2: Data Repository Deposit

Reviewer Comment: Consider depositing your raw data in a repository to increase citation advantage.

Response: Thank you for your suggestion. Unfortunately, I must decline due to the sensitive nature of the data, which involves the privacy of individuals.

Point 3: Grant Information

Reviewer Comment: Ensure the grant information in the ‘Funding Information’ and ‘Financial Disclosure’ sections match.

Response: Thank you for your comment. The project "Medical Science Research Project Plan" from the Hebei Provincial Health Commission does not appear in the official system and does not provide financial support. Therefore, there are no discrepancies between the ‘Funding Information’ and ‘Financial Disclosure’ sections as no grant funding was received for this study.

Point 4: Data Availability Statement

Reviewer Comment: Confirm whether your submission contains all raw data required to replicate the results of your study.

Response: We have uploaded the original data as supplementary material to facilitate the replication of our study.

Point 5: Author Amendments

Reviewer Comment: Include authors Lianxia Geng, Shulin Jiang, Chuanjie Yang, Kaiyue Xu, Shufang Shen, and Wenwen Cao.

Response: We have included the mentioned authors in the manuscript submission data, reflecting their contributions accurately.

Point 6: Ethics Statement

Reviewer Comment: Include your full ethics statement in the ‘Methods’ section.

Response: The full ethics statement is already included in the ‘Methods’ section. The study was approved by the Ethics Committee of the Second Hospital of Hebei Medical University (Grant number: 2022-R137).

Point 7: Reference List Review

Reviewer Comment: Review your reference list to ensure it is complete and correct.

Response: We reviewed and updated our reference list using EndNote. Deleted references are marked in red and crossed out, while new references are highlighted in yellow.

Response to Reviewer #1,

Thank you for your insightful review and constructive feedback on our manuscript titled "Public Awareness of Gastric Cancer Risk Factors and Screening Behaviours in Shijiazhuang, China: A Community-based Survey."

We appreciate your recognition of the importance of our study and your positive comments on its design and analysis. We have addressed all of these comments, and we highlighted all the revision in yellow colour.

Comment 1：

It seems that the main interest of authors is to find the link between public knowledge and screening. It could be good to emphasize also how improving public knowledge could lead to change of lifestyle and acting as a tertiary preventive measure.

Response:

We have revised the text to address your concerns. The details as follows:

1 We added the following sentence in the Introduction:

”Unfortunately, research from many countries shows that adherence in the general population is worryingly low. Effective prevention necessitates both collective actions and individual actions . A comprehensive understanding of public awareness regarding GC risk factors significantly influences the development and execution of effective prevention strategies against this disease. For individual actions to occur, public awareness of cancer risk factors is a crucial prerequisite. ”(Page4, Line79-85)

“The findings have significant implications for health-related behavior change and may enhance GC prevention and control efforts.”(Page6, Line124-126)

2 We also change [it is essential to develop effective communication strategies that highlight how obesity increases an individual’s risk of cancer. Enhancing awareness in this regard could improve the efficacy of interventions and messages aimed at promoting weight loss and prevention of weight gain. These efforts are crucial in the present time to address the rising obesity rates and combat the associated risks of cancer.] to [Therefore, an effective method is required for governments, social media, professionals and the community to work together to improve cancer awareness and modify unhealthy lifestyles. Leveraging the internet to disseminate professional and authoritative prevention knowledge can effectively promote a healthy lifestyle and foster a supportive environment for it. By enhancing public awareness, compliance with health behaviors will subsequently improve, enabling more individuals to adopt and maintain practices that are beneficial to their well-being.(Page16, Line303-310)]in the Discussion.

Comment 2：

Figures look like being screenshot with low resolution. It is important to have high quality images for the final publication.

Response:

We acknowledge your concern regarding the quality of the figures. We have replaced all figures with high-resolution versions to ensure clarity and quality for the final publication.

We believe these changes address your concerns and enhance the overall quality of our manuscript. We have attached the revised manuscript for your review.

Thank you once again for your constructive comments and for helping us improve our work.

Response to Reviewer #2,

Thank you very much for your positive feedback and valuable suggestions on our manuscript titled "Public Awareness of Gastric Cancer Risk Factors and Screening Behaviours in Shijiazhuang, China: A Community-based Survey." We greatly appreciate your time and effort in reviewing our work. We are especially grateful for your kind words regarding the quality and importance of our study to the scientific community.

We have carefully considered your suggestions and made the following revisions to improve the manuscript. In the revised manuscript, all the changes have been highlighted in yellow.

Comment 1:

Present an international overview of the theme in the introduction (what was done similarly in other countries, or even what was not done)

Response:

We have expanded the Introduction to provide an international perspective on the topic. Please see Page 4 , lines 77-79, Page4-5, lines 85-92, and Page 5-6, lines 102-112.

Comment 2:

Improve the practical implications of the study (How important is the study for society? How important is the study for professionals? How important is the study for the health system? Etc..

Response:

1 We have added another paragraph in Discussion.

“Our findings have important clinical and public health implications. First, we provide valuable reference data to help decision-makers enhance cancer awareness and screening rates. It is advisable for them to establish an authoritative platform for science communication, to compile and disseminate core information and key points on cancer prevention and treatment. Simultaneously, they should optimize cancer screening management and establish a tiered screening system. Secondly, professionals will tailor health education initiatives to address public misconceptions and knowledge gaps, thereby tackling inequalities in awareness. Additionally, barriers to screening motivate doctors to develop novel, non-invasive, and precise biomarkers, which further advance screening technologies. Third, although raising public awareness is a relatively gentle method for achieving behavioral change, it is immensely beneficial for enhancing the overall health literacy of the population and encouraging long-term adherence to healthy behaviors. Better understanding of risk factors and increased prevention efforts offer cost-effective and sustainable means to reduce the global cancer burden in the longer term.”(Page 21-22, Lines 426-440)

2 We have made minor adjustments in other parts of the Discussion to ensure consistency in tone and completeness. Please see Page 17, lines 339-344, and Page 21, lines 417-422.

Once again, we sincerely appreciate your constructive feedback. Your suggestions have undoubtedly helped improve the quality and clarity of our manuscript. We look forward to your further comments and hope that our revised manuscript meets your expectations.

Thank you for your continued support.

---

## [Decision Letter · Decision Letter 1]

20 Sep 2024

Public Awareness of Gastric Cancer Risk Factors and Screening Behaviours in Shijiazhuang, China: A Community-based Survey

PONE-D-23-38363R1

Dear Dr. zhao,

We’re pleased to inform you that your manuscript has been judged scientifically suitable for publication and will be formally accepted for publication once it meets all outstanding technical requirements.

Kind regards,

Pengpeng Ye

Academic Editor

PLOS ONE

Additional Editor Comments (optional):

Reviewers' comments:

Reviewer's Responses to Questions

**Comments to the Author**

1. If the authors have adequately addressed your comments raised in a previous round of review and you feel that this manuscript is now acceptable for publication, you may indicate that here to bypass the “Comments to the Author” section, enter your conflict of interest statement in the “Confidential to Editor” section, and submit your "Accept" recommendation.

Reviewer #1: All comments have been addressed

Reviewer #2: All comments have been addressed

Reviewer #3: (No Response)

2. Is the manuscript technically sound, and do the data support the conclusions?

Reviewer #1: Yes

Reviewer #2: Yes

Reviewer #3: Yes

3. Has the statistical analysis been performed appropriately and rigorously? 

Reviewer #1: Yes

Reviewer #2: Yes

Reviewer #3: Yes

4. Have the authors made all data underlying the findings in their manuscript fully available?

Reviewer #1: No

Reviewer #2: Yes

Reviewer #3: Yes

5. Is the manuscript presented in an intelligible fashion and written in standard English?

Reviewer #1: Yes

Reviewer #2: Yes

Reviewer #3: Yes

6. Review Comments to the Author

Reviewer #1: Thanks for improving the manuscript and answering the comments. Hopefully this work will be continued to provide social awareness and appropriate policy settings. I have no more comments.

Reviewer #2: The authors have made all the adjustments requested in the initial review with accuracy and diligence. Their modifications demonstrated a remarkable commitment to the quality of the work, addressing all issues raised by the reviewers comprehensively and satisfactorily. Their responses to suggestions and criticisms were prompt and well-founded, resulting in significant improvements to the text. I am pleased with the effort and attention to detail demonstrated by the authors during this review process, and my recommendation is to accept the article.

Reviewer #3: (No Response)

7. PLOS authors have the option to publish the peer review history of their article (what does this mean?). If published, this will include your full peer review and any attached files.

Reviewer #1: **Yes: **Behnoush Abedi-Ardekani

Reviewer #2: **Yes: **Mateus Antunes

Reviewer #3: No

---

## [Editor Report · Acceptance letter]

27 Sep 2024

PONE-D-23-38363R1 

PLOS ONE

Dear Dr. Zhao, 

I'm pleased to inform you that your manuscript has been deemed suitable for publication in PLOS ONE. Congratulations! Your manuscript is now being handed over to our production team.

Kind regards, 

on behalf of

Dr. Pengpeng Ye 

Academic Editor

PLOS ONE